# IMED-RL: Regret optimal learning of ergodic Markov decision processes

**Fabien Pesquerel**[*]
fabien.pesquerel@inria.fr

**Odalric-Ambrym Maillard**
odalric.maillard@inria.fr

Univ. Lille, CNRS, Inria, Centrale Lille, UMR 9198-CRIStAL, F-59000 Lille, France

## Abstract

We consider reinforcement learning in a discrete, undiscounted, infinite-horizon Markov Decision Problem (MDP) under the average reward criterion, and focus on the minimization of the regret with respect to an optimal policy, when the learner does not know the rewards nor the transitions of the MDP. In light of their success at regret minimization in multi-armed bandits, popular bandit strategies, such as the optimistic `UCB`, `KL-UCB` or the Bayesian Thompson sampling strategy, have been extended to the MDP setup. Despite some key successes, existing strategies for solving this problem either fail to be provably asymptotically optimal, or suffer from prohibitive burn-in phase and computational complexity when implemented in practice. In this work, we shed a novel light on regret minimization strategies, by extending to reinforcement learning the computationally appealing Indexed Minimum Empirical Divergence (`IMED`) bandit algorithm. Traditional asymptotic problem-dependent lower bounds on the regret are known under the assumption that the MDP is *ergodic*. Under this assumption, we introduce `IMED-RL` and prove that its regret upper bound asymptotically matches the regret lower bound. We discuss both the case when the supports of transitions are unknown, and the more informative but a priori harder-to-exploit-optimally case when they are known. Rewards are assumed light-tailed, semi-bounded from above. Last, we provide numerical illustrations on classical tabular MDPs, *ergodic* and *communicating* only, showing the competitiveness of `IMED-RL` in finite-time against state-of-the-art algorithms. `IMED-RL` also benefits from a light complexity.

## 1 Introduction

We study Reinforcement Learning (RL) with an unknown finite Markov Decision Problem (MDP) under the average-reward criterion in which a learning algorithm interacts sequentially with the dynamical system, without any reset, in a single and infinite sequence of observations, actions, and rewards while trying to maximize its total accumulated rewards over time. Formally, we consider a finite MDP $\mathbf{M} = (\mathcal{S}, \mathcal{A}, \mathbf{p}, \mathbf{r})$ where $\mathcal{S}$ is the finite set of states, $\mathcal{A} = (\mathcal{A}_s)_{s \in \mathcal{S}}$ specifies the set of actions available in each state and we introduce the set of pairs $\mathcal{X}_{\mathbf{M}} = \{(s, a) : s \in \mathcal{S}, a \in \mathcal{A}_s\}$ for convenience. Further[1], $\mathbf{p} : \mathcal{X}_{\mathbf{M}} \to \mathcal{P}(\mathcal{S})$ is the transition distribution function and $\mathbf{r} : \mathcal{X}_{\mathbf{M}} \to \mathcal{P}(\mathbb{R})$ the reward distribution function, with corresponding mean reward function denoted by $\mathbf{m} : \mathcal{X}_{\mathbf{M}} \to \mathbb{R}$. An agent interacts with the MDP at discrete time steps $t \in \mathbb{N}^*$ and yields a random sequence $(s_t, a_t, r_t)_t$ of states, actions, and rewards in the following way. At each time step $t$, the agent observes the current state $s_t$ and decides the action $a_t$ to take based on $s_t$ and possibly past information, *i.e.* previous elements of the sequence. After playing $a_t$, it observes a reward $r_t \sim \mathbf{r}(s_t, a_t)$, the current state of the MDP changes to $s_{t+1} \sim \mathbf{p}(\cdot|s_t, a_t)$ and the agent proceeds sequentially. In the *average-*

---

[1]Given a set $E$, $\mathcal{P}(E)$ denotes the set of probability distributions on $E$.

36th Conference on Neural Information Processing Systems (NeurIPS 2022).

*reward setting*, one is interested in maximizing the limit, $\frac{1}{T}\sum_{t=1}^{T} r_t$, when $T \to \infty$, providing it exists. This setting is a popular framework for studying sequential decision making problems; it can be traced back to seminal papers such as those of Graves and Lai [1997] and Burnetas and Katehakis [1997] This theoretical framework allows to study the *exploration-exploitation* trade-off that arises from the sequential optimization problem a learner is trying to solve while being uncertain about the very problem it is optimizing.

In this paper, one is interested in developing a sampling strategy that is *optimal* amongst strategies that aim at maximizing the average-reward, *i.e.* balancing exploration and exploitation in an optimal way. To assert optimality, we define the notion of *regret* and state a *regret lower bound* with the purpose of defining a theoretically sound notion of optimality that is *problem-dependent*. While *regret* defines the discrepancy to optimality of a learning strategy, a *problem-dependent regret lower bound* will formally assess the minimal regret that any learning algorithm must incur on a given MDP problem by computing a minimal rate of exploration. Because this minimal rate of exploration depends on the problem, it is said to be problem-dependent, as opposed to worst case regret study that can exist in the MDP literature (*e.g.* Jaksch et al. [2010]). Regret lower bounds currently exist in the literature when the MDP **M** is assumed to be *ergodic*[2]. Hence we hereafter make this assumption, in order to be able to compare the regret of our algorithm to an optimal bound. Similarly, to ensure fast enough convergence of the empirical estimate of the reward to the true mean, an assumption controlling the rate of convergence to the mean is necessary.

**Assumption 1** (Light-tail rewards). *For all $x \in \mathcal{X}_\mathbf{M}$, the moment generating function of the reward exists in a neighborhood of* $0$*:* $\exists \lambda_x > 0, \forall \lambda \in \mathbb{R}$ *such that* $|\lambda| < \lambda_x, \mathbb{E}_{R \sim \mathbf{r}(x)}[\exp(\lambda R)] < \infty$.

**Policy** Regret and ergodicity are defined using properties of the set of stationary deterministic policies $\Pi(\mathbf{M})$ on $\mathbf{M}$. On $\mathbf{M}$, each stationary deterministic policy $\pi : \mathcal{S} \to \mathcal{A}_s$ defines a Markov reward process, *i.e.* a Markov chain on $\mathcal{S}$ with kernel $\mathbf{p}_\pi : s \in \mathcal{S} \mapsto \mathbf{p}(\cdot|s, \pi(s)) \in \mathcal{P}(\mathcal{S})$ together with rewards $\mathbf{r}_\pi : s \in \mathcal{S} \mapsto \mathbf{r}(s, \pi(s)) \in \mathcal{P}(\mathbb{R})$ and associated mean rewards $\mathbf{m}_\pi : s \in \mathcal{S} \mapsto \mathbf{m}(s, \pi(s)) \in \mathbb{R}$. The $t$-steps transition kernel of $\pi$ on $\mathbf{M}$ is denoted $\mathbf{p}_\pi^t$. We denote $\overline{\mathbf{p}}_\pi = \lim_{T \to \infty} \frac{1}{T} \sum_{t=1}^{T} \mathbf{p}_\pi^{t-1} : \mathcal{S} \to \mathcal{P}(\mathcal{S})$ the Cesaro-average of $\mathbf{p}_\pi$. A learning agent is executing a sequence of policies $\pi_t \in \Pi(\mathbf{M})$, $t \geqslant 1$, where $\pi_t$ depends on past information $(s_{t'}, a_{t'}, r_{t'})_{t' < t}$. With a slight abuse of notation, a sequence of identical decision rules, $\pi_t = \pi$ for all $t$, is also denoted $\pi$.

**Gain** The cumulative reward (value) at time $T$, starting from an initial state $s_1$ of policy $\pi = (\pi_t)_t$ is formally given by

$$V_{s_1}(\mathbf{M}, \pi, T) = \mathbb{E}_{\pi, \mathbf{M}, s_1}\left[\sum_{t=1}^{T} r_t\right] = \mathbb{E}_{\pi, \mathbf{M}, s_1}\left[\sum_{t=1}^{T} \mathbf{m}(s_t, a_t)\right] = \sum_{t=1}^{T}\left(\prod_{t'=1}^{t-1} \mathbf{p}_{\pi_{t'}} \mathbf{m}_{\pi_{t'}}\right)(s_1). \quad (1)$$

For $\pi \in \Pi(\mathbf{M})$, the average-reward $\frac{1}{T} V_{s_1}(\mathbf{M}, \pi, T)$ tends to $(\overline{\mathbf{p}}_\pi \mathbf{m})(s_1)$ as $T \to \infty$. The gain of policy $\pi \in \Pi(\mathbf{M})$, when starting from state $s_1$ is defined by $\mathbf{g}_\pi(s_1) = (\overline{\mathbf{p}}_\pi \mathbf{m})(s_1)$ and the optimal gain is defined as $\mathbf{g}^\star(s_1) = \max_{\pi \in \Pi(\mathbf{M})} \mathbf{g}_\pi(s_1)$. $\mathcal{O}_s(\mathbf{M}) = \{\pi \in \Pi : \mathbf{g}_\pi(s) = \mathbf{g}^\star(s)\}$ is the set of policies achieving maximal gain on $\mathbf{M}$ starting from state $s$.

**Definition 1** (Regret). *The regret at time $T$ of a learning policy $\pi = (\pi_t)_t$ starting at state $s$ on an MDP $\mathbf{M}$ is defined with respect to any $\pi^\star \in \mathcal{O}_s(\mathbf{M})$, as*

$$\mathcal{R}_{\pi, s}(\mathbf{M}, T; \pi^\star) = V_s(\mathbf{M}, \pi^\star, T) - V_s(\mathbf{M}, \pi, T). \quad (2)$$

In this paper, we aim to find a learning algorithm with *asymptotic* minimal regret. The Lemma 1 will prove that for all optimal policies, $\pi^\star$, regrets are the same up to a bounded term that therefore does not count in asymptotic analysis. Some authors such as Bourel et al. [2020] define the regret as $T\mathbf{g}^\mathbf{M}(s) - V_s(\mathbf{M}, \pi, T)$ which is equal to the one we defined up to a bounded term (again by Lemma 1). No stationary policy can be optimal at all time and the important fact is that all those notions of regret induce the same asymptotic lower bound.

In the considered setting, the learning agent interacts with the MDP without any reset. The minimal assumption would be to allow the agent to come back with positive probability from any initial

---

[2]We prefer the term *ergodic* over the more accurate one, *irreducible* as it is a standard abuse of terminology in the MDP community. Mathematically, an MDP is ergodic if both irreducible, aperiodic and positive recurrent.

mistake in finite time, so that the agent is not stuck in a sub-optimal area of the system. This is assuming that the MDP is *communicating*, that is $\forall s, s', \exists \pi, t \in \mathbb{N} : \mathbf{p}_\pi^t(s'|s) > 0$. However, in the literature, lower bounds on the regret are stated for MDPs satisfying a stronger assumption, *ergodicity*. Since one is interested in crafting an algorithm matching a lower bound, we consider this stronger assumption.

**Assumption 2** (Ergodic MDP). *The MDP* $\mathbf{M}$ *is ergodic, that is* $\forall s, s', \forall \pi, \exists t \in \mathbb{N} : \mathbf{p}_\pi^t(s'|s) > 0$.

Intuitively, this means that for all policies and all couples of states, there exists a finite trajectory of positive probability between the states. Interestingly, the ergodic property can be assumed on the MDP or on the set of policies in which we seek an optimal one. For instance, in any communicating MDP all $\varepsilon$-soft policies[3] are ergodic; more in the Experiment section 5 and Appendix E.

**Related work**   Had the MDP only one state, it would be a bandit problem. Lower bound on the bandit regret and algorithms matching this lower bound, sometimes up to a constant factor, are well studied in the bandit literature. Therefore, bandit sampling strategies with known theoretical guarantees have inspired RL algorithms. The `KL-UCB` algorithm (Burnetas and Katehakis [1996], Maillard et al. [2011]), has inspired the strategy of the seminal paper of Burnetas and Katehakis [1997], as well the more recent `KL-UCRL` strategy (Filippi et al. [2010] Talebi and Maillard [2018]). Inspired by the `UCB` algorithm (Agrawal [1995], Auer et al. [2002]), a number of strategies implementing the optimism principle have emerged such as `UCRL` (Auer and Ortner [2006]), `UCRL2` (Jaksch et al. [2010]) and `UCRL3` (Bourel et al. [2020] (and beyond, Azar et al. [2017], Dann et al. [2017] for the related episodic setup). The strategy `PSRL` (Osband et al. [2013]) is inspired by Thompson sampling (Thompson [1933]).

**Outline and contribution**   In this work, we build on the `IMED` strategy (Honda and Takemura [2015]), a bandit algorithm that benefits from practical and optimal guarantees but has never been used by the RL community. We fill this gap by proposing the `IMED-RL` algorithm which we prove to be asymptotically optimal for the average-reward criterion. We revisit the notion of skeleton (Equation 12) introduced in the seminal work of Burnetas and Katehakis [1997], with a subtle but key modification that prevents a prohibitive burn-in phase (see Appendix G for further details). Further, this novel notion of skeleton enables `IMED-RL` to remove any tracking or hyperparameter and mimic a *stochastic-policy-iteration-like* algorithm. [4] Further, this skeleton scales naturally with the studied MDP as it does not explicitly refer to absolute quantities such as the time. We prove that our proposed `IMED-RL` is asymptotically optimal and show its numerical competitivity.

Building on `IMED`, we make an additional assumption on the reward that is less restrictive than the common bounded reward hypothesis made in the RL community.

**Assumption 3** (Semi-bounded rewards). *For all* $x \in \mathcal{X}$, $r(x)$ *belongs to a subset* $\mathcal{F}_x \subset \mathcal{P}(\mathbb{R})$ *known to the learner*.[5] *There exists a known quantity* $m_{\max}(x) \in \mathbb{R}$ *such that for all* $x \in \mathcal{X}$, *the support* $\mathtt{Supp}(\mathbf{r}(x))$ *of the reward distribution is semi-bounded from above,* $\mathtt{Supp}(\mathbf{r}(x)) \subset ]-\infty, m_{max}(x)]$, *and its mean satisfies* $\mathbf{m}(x) < m_{\max}(x)$.

**Ergodic assumption**   While many recent works focused on worst-case regret bounds only (e.g. Domingues et al. [2021], Zanette and Brunskill [2019], Jin et al. [2018] and citations therein), studying problem-dependent optimal regret bounds has been somewhat overlooked. Being more general is always more appealing but the restriction from communicating MDPs to ergodic MDPs allows us to target exact asymptotic optimality ; not just bound, not just worst-case bound. Ergodic MDPs is the only case in which explicit problem-dependent lower bounds are known and hence can be directly used to build a strategy. Indeed, the main challenge towards problem-dependent optimality is that existing lower bounds for exploration problems in MDPs are usually written in terms of non-convex optimization problems. This *implicit* form makes it hard to understand the actual complexity of the setting and, thus, to design optimal algorithms. Existing proof strategies for state-of-the-art algorithms (`UCRL`, `PSRL`, *etc*) ensure a regret for communicating MDPs but fail to provide optimality guarantees even in the ergodic case. We believe that deriving a sharp result in the ergodic case

---

[3]A policy $\pi : \mathcal{S} \to \mathcal{P}(\mathcal{A}_s)$ is $\varepsilon$-soft if $\pi(a|s) \geqslant \varepsilon/|\mathcal{A}_s|$ for all $s$ and $a$.

[4]The skeleton in Burnetas and Katehakis [1997] is sometimes empty at some states, when $t$ is too small, this causes the strategy to work well only after $t$ is large enough to ensure that the skeleton contains at least one action in each state.

[5]*e.g.* Bernoulli, multinomial with unknown support, beta, truncated Gaussians, a mixture of those, *etc*.

might prove to be insightful to pave the way towards the communicating case. From a theoretical standpoint, related to `UCRL` type strategy, modern analysis of `KL-UCRL` by Talebi and Maillard [2018] also makes the ergodic assumption. This hypothesis has also been used in the theoretical work of Tewari and Bartlett [2007] and the work of Ok et al. [2018] that concerns structured MDPs. Related to this assumption are works that are interested in identification and sample complexity. Wang [2017] introduced a primal-dual method to compute an $\varepsilon$-optimal policy and bound the number of sample transitions to reach this goal. Jin and Sidford [2020] relaxed the ergodic hypothesis by using a mixing hypothesis that implies the uniqueness of recurrent class for each policy. In this setting, the authors also derive a bound on the number of samples to compute an $\varepsilon$-optimal policy.

## 2 Regret lower bound

In this section, we recall the regret lower bound for ergodic MDPs and provide a few insights about it.

**Characterizing optimal policies** Relying on classical results that can be found in the books of Puterman [1994] and Hernández-Lerma and Lasserre [1996], we give a useful characterization of optimal policies that is used to derive a regret lower bound. Under the ergodic Assumption 2 of MDP **M**, for all policy $\pi \in \Pi(\mathbf{M})$, the gain is independent from the initial state, *i.e.* $\mathbf{g}_\pi(s) = \mathbf{g}_\pi(s')$ for all states $s$ and $s'$, and we denote it $\mathbf{g}_\pi$. Similarly, the set of optimal policies $\mathcal{O}(\mathbf{M})$ is state-independent since $\mathcal{O}_s(\mathbf{M}) = \mathcal{O}_{s'}(\mathbf{M})$. Any policy $\pi$ satisfy the following fixed point property

$$\text{(Poisson equation)} \qquad \mathbf{g}_\pi + \mathbf{b}_\pi(s) = \mathbf{m}_\pi(s) + (\mathbf{p}_\pi \mathbf{b}_\pi)(s), \qquad (3)$$

where $\mathbf{b}_\pi : \mathcal{S} \to \mathbb{R}$ is called the bias function and is defined up to an additive constant by $\mathbf{b}_\pi(s) = \left( \sum_{t=1}^\infty (\mathbf{p}_\pi^{t-1} - \overline{\mathbf{p}}_\pi) \mathbf{m}_\pi \right)(s)$. We highlight that bias plays a role similar to the value function in the discounted reward setting in which the gain is always zero and Equation 3 reduces to the Bellman equation, giving a direction in which extend our results to this other RL setting. Interestingly, for any communicating and a fortiori ergodic MDP, the span $\mathbb{S}(\mathbf{b}_\pi) = \max_{s \in \mathcal{S}} \mathbf{b}_\pi(s) - \min_{s \in \mathcal{S}} \mathbf{b}_\pi(s)$ of the bias function of any policy is bounded, which allows to decompose the regret in the useful following way.

**Lemma 1** (Regret decomposition). *Under the ergodic assumption 2, for all optimal policy $\star \in \mathcal{O}(\mathbf{M})$, the regret of any policy $\pi = (\pi_t)_t$ can be decomposed as*

$$\mathcal{R}_{\pi,s_1}(\mathbf{M}, T; \star) = \sum_{x \in \mathcal{X}_\mathbf{M}} \mathbb{E}_{\pi,s_1}[N_x(T)] \Delta_x(\mathbf{M}) + \underbrace{\left( \left[ \prod_{t=1}^T \mathbf{p}_{\pi_t} - \mathbf{p}_\star^t \right] b_\star \right)(s_1)}_{\leqslant \mathbb{S}(\mathbf{b}_\star)}, \qquad (4)$$

*where $N_{s,a}(T) = \sum_{t=1}^T \mathbb{1}\{s_t = s, a_t = a\}$ counts the number of time the state-action pair $(s,a)$ has been sampled and $\Delta_{s,a}(\mathbf{M})$ is the sub-optimality gap of the state-action pair $(s,a)$ in **M**,*

$$\Delta_{s,a}(\mathbf{M}) = \mathbf{m}(s,a) + \mathbf{p}_a \mathbf{b}_\star(s) - \mathbf{m}_\star(s) - \mathbf{p}_\star \mathbf{b}_\star(s) = \mathbf{m}(s,a) + \mathbf{p}_a \mathbf{b}_\star(s) - \mathbf{g}_\star - \mathbf{b}_\star(s) \quad (5)$$

*with $\mathbf{p}_a = \mathbf{p}(\cdot|s,a)$ by a slight abuse of notation. Action $a \in \mathcal{A}_s$ is optimal if and only if $\Delta_{s,a}(\mathbf{M}) = 0$, otherwise, it is said sub-optimal.*

This result can be found in Puterman [1994] and is rederived in Appendix C.

Under the ergodic Assumption 2 of MDP **M**, all optimal policies satisfy a Poisson equation while some are also being characterized by the optimal Poisson equation (see Hernández-Lerma and Lasserre [1996]), used to compute the optimal gain and a bias function associated to an optimal policy,

$$\mathbf{g}^\mathbf{M} + \mathbf{b}^\mathbf{M}(s) = \max_{a \in \mathcal{A}_s} \left\{ \mathbf{m}(s,a) + \sum_{s' \in \mathcal{S}} \mathbf{p}(s'|s,a) \mathbf{b}^\mathbf{M}(s') \right\}. \qquad (6)$$

**Lower bound** To assess the minimal sampling complexity of a sub-optimal state action pair, one must compute how far a sub-optimal state-action pair is from being optimal from an information point-of-view. A sub-optimal state-action pair $(s,a) \in \mathcal{X}_\mathbf{M}$ is said to be *critical* if it can be made optimal by changing reward $\mathbf{r}(s,a)$ and transition $\mathbf{p}(\cdot|s,a)$ while respecting the assumptions on the rewards and transitions. Formally, let $\varphi_\mathbf{M} : \mathcal{P}(\mathbb{R} \times \mathcal{S}) \to \mathbb{R}$,

$$\varphi_\mathbf{M}(\nu \otimes q) = \mathbb{E}_{R \sim \nu}[R] + q \mathbf{b}^\mathbf{M} \qquad (7)$$

denotes the potential function of $\nu \otimes q$ in $\mathbf{M}$, where $\nu \otimes q$ is the product measure of $\nu$ and $q$. A pair $(s, a) \in \mathcal{X}_\mathbf{M}$ is *critical* if it is sub-optimal and there exists $\nu \in \mathcal{F}_{s,a}$ and $q \in \mathcal{P}(\mathcal{S})$ such that

$$\varphi_\mathbf{M}(\nu \otimes q) > \gamma_s(\mathbf{M}) \quad \text{where } \gamma_s(\mathbf{M}) \stackrel{\text{def}}{=} \mathbf{g}^\mathbf{M} + \mathbf{b}^\mathbf{M}(s). \tag{8}$$

Note that $\gamma_s(\mathbf{M}) = \max\limits_{a \in \mathcal{A}_s} \varphi_\mathbf{M}(\mathbf{r}(s, a) \otimes \mathbf{p}(s, a))$ by the optimal Poisson equation (6).

**Definition 2** (Sub-optimality cost). *The **sub-optimality cost** of a sub-optimal state-action pair* $(s, a) \in \mathcal{X}_\mathbf{M}$ *is defined as* $\underline{\mathbf{K}}_{s,a}(\mathbf{M}) \stackrel{\text{def}}{=} \underline{\mathbf{K}}_{s,a}(\mathbf{M}, \gamma_s(\mathbf{M}))$ *where*

$$\underline{\mathbf{K}}_{s,a}(\mathbf{M}, \gamma) = \inf\limits_{\substack{\nu \in \mathcal{F}_{s,a} \\ q \in \mathcal{P}(\mathcal{S})}} \{\mathrm{KL}(\mathbf{r}(s, a) \otimes \mathbf{p}(\cdot|s, a), \nu \otimes q) \; : \; \varphi_\mathbf{M}(\nu \otimes q) > \gamma\}, \tag{9}$$

*and* KL *denotes the Kullback-Leibler divergence between distributions.*

A lower bound on the regret may now be stated for a certain class of learner, the set of uniformly consistent learning algorithm, *i.e.* those policies $\pi = (\pi_t)_t$ such that $\mathbb{E}_{\pi,\mathbf{M}}(N_{s,a}(T)) = o(T^\alpha)$ for all sub-optimal state-action pair $(s, a)$ and $0 < \alpha < 1$ (see Agrawal et al. [1989]).

**Theorem 1** (Regret lower bound Burnetas and Katehakis [1997]). *Let* $\mathbf{M} = (\mathcal{S}, \mathcal{A}, \mathbf{p}, \mathbf{r})$ *be an MDP satisfying Assumptions 1, 2, 3. For all uniformly consistent learning algorithm* $\pi$,

$$\liminf\limits_{T \to \infty} \frac{\mathbb{E}_{\pi,\mathbf{M}}[N_{s,a}(T)]}{\log T} \geqslant \frac{1}{\underline{\mathbf{K}}_{s,a}(\mathbf{M})} \tag{10}$$

*with the convention that* $1/\infty = 0$. *The regret lower bound is*

$$\liminf\limits_{T \to \infty} \frac{\mathcal{R}_\pi(\mathbf{M}, T)}{\log T} \geqslant \sum\limits_{(s,a) \in \mathcal{C}(\mathbf{M})} \frac{\Delta_{s,a}(\mathbf{M})}{\underline{\mathbf{K}}_{s,a}(\mathbf{M})} \tag{11}$$

*where* $\mathcal{C}(\mathbf{M}) = \{(s, a) : 0 < \underline{\mathbf{K}}_{s,a}(\mathbf{M}) < \infty\}$ *is called the set of critical state-action pairs. Those are the state-action pairs* $(s, a)$ *that could be confused for an optimal one if we were to change their associated rewards and transitions distributions at the displacement cost of* $\underline{\mathbf{K}}_{s,a}(\mathbf{M})$.

## 3 The `IMED-RL` Algorithm

In this section we introduce and detail the `IMED-RL` algorithm, whose regret matches this fundamental lower bound and extends the `IMED` strategy from Honda and Takemura [2015] to ergodic MDPs. Indeed, for a single-state MDP, that is a multi-armed bandit, `IMED-RL` simply reduces to `IMED`.

**Empirical quantities** `IMED-RL` is a *model-based* algorithm that keeps empirical estimates of the transitions $\mathbf{p}$ and rewards $\mathbf{r}$ as opposed to *model-free* algorithm such as Q-learning. We denote by $\hat{\mathbf{r}}_t(s, a) = \hat{\mathbf{r}}(s, a; N_{s,a}(t))$ and $\hat{\mathbf{p}}_t(s, a) = \hat{\mathbf{p}}(s, a; N_{s,a}(t))$ the empirical reward distributions and transition vectors after $t$ time steps, *i.e.* using $N_{s,a}(t)$ samples from the distribution $\mathbf{r}(s, a)$. Initially, $\hat{\mathbf{p}}(s, a; 0)$ is the uniform probability over the state space and $\hat{\mathbf{p}}(s, a; k) = (1 - 1/k)\hat{\mathbf{p}}(s, a; k - 1) + (1/k)\mathbf{s}_k$, where $\mathbf{s}_k$ is a vector of zeros except for a one at index $s_k$, the $k^{th}$ samples drawn from $\mathbf{p}(\cdot|s, a)$. This defines at each time step $t$ an empirical MDP $\widehat{\mathbf{M}}_t = (\mathcal{S}, \mathcal{A}, \hat{\mathbf{p}}_t, \hat{\mathbf{r}}_t)$. On this empirical MDP, for each state, some actions have been sampled more than others and their empirical quantities are therefore better estimated. We call *skeleton* at time $t$ the subset of state-action pairs that can be considered sampled enough at time $t$; it is defined by restricting $\mathcal{A}_s$ to $\mathcal{A}_s(t)$ for all state $s \in \mathcal{S}$, with

$$\mathcal{A}_s(t) = \left\{ a \in \mathcal{A}_s \; : \; N_{s,a}(t) \geqslant \log^2\left(\max\limits_{a' \in \mathcal{A}_s} N_{sa'}(t)\right) \right\}. \tag{12}$$

Since $x > \log^2 x$, $\mathcal{A}_s(t) \neq \emptyset$, hence $\mathcal{A}(t) = (\mathcal{A}_s(t))_s$ contains at least one deterministic policy. We note that the MDP $\mathbf{M}(\mathcal{A}(t)) \stackrel{\text{def}}{=} (\mathcal{S}, \mathcal{A}(t), \mathbf{p}, \mathbf{r})$ defined by restricting the set of actions to $\mathcal{A}(t) \subseteq \mathcal{A}$ is an ergodic MDP. The restricted empirical MDP $\widehat{\mathbf{M}}_t(\mathcal{A}(t)) \stackrel{\text{def}}{=} (\mathcal{S}, \mathcal{A}(t), \hat{\mathbf{p}}_t, \hat{\mathbf{r}}_t)$ also is ergodic thanks to the ergodic initialization of the estimate $\hat{\mathbf{p}}$. Inspired by `IMED`, we define the `IMED-RL` index.

**Definition 3** (`IMED-RL` index). *For all state-action pairs* $(s, a) \in \mathcal{X}_\mathbf{M}$, *let us define* $\mathbf{K}_{s,a}(t) \stackrel{\text{def}}{=} \underline{\mathbf{K}}_{s,a}\left(\widehat{\mathbf{M}}_t(\mathcal{A}(t)), \hat{\gamma}_s(t)\right)$ *with empirical threshold* $\hat{\gamma}_s(t) \stackrel{\text{def}}{=} \max\limits_{a \in \mathcal{A}_s} \varphi_{\widehat{\mathbf{M}}_t(\mathcal{A}(t))}(\hat{\mathbf{r}}(s, a) \otimes \hat{\mathbf{p}}(s, a))$ *Then, the* `IMED-RL` *index of* $(s, a)$ *at time* $t$, $\mathbf{H}_{s,a}(t)$, *is defined as*

$$\mathbf{H}_{s,a}(t) = N_{s,a}(t)\mathbf{K}_{s,a}(t) + \log N_{s,a}(t). \tag{13}$$

Note that $\hat{\gamma}_s(t) \neq \gamma_s(\hat{\mathbf{M}}_t(\mathcal{A}(t)))$ as the maximum is taken over all $a \in \mathcal{A}_s$ an not just $a \in \mathcal{A}_s(t)$.

**Known support of transitions**  Were the support of transition known, the infimum in sub-optimality cost $\underline{\mathbf{K}}_{s,a}$ defined by Equation 9 would be redefined as one over the set $\{q \in \mathcal{P}(\mathcal{S}) : \mathrm{Supp}(q) = \mathrm{Supp}(\mathbf{p}(\cdot|s,a))\}$, modifying both the lower bound and IMED-RL index.

IMED-RL **algorithm**  The IMED-RL algorithm consists in playing at each time step $t$, an action $a_t$ of minimal IMED-RL index at the current state $s_t$. The intuition behind the IMED-RL index is similar to the one of the IMED index for bandits and stems from an information theoretic point-of-view of the lower bound. At a given time $t$, the frequency of play $\frac{N_{s,a}(t)}{N_s(t)}$ of action $a \in \mathcal{A}_s$ in state $s \in \mathcal{S}$, should be larger than or equal to its posterior probability of being the optimal action in that state, $\exp(-N_{s,a}(t)\mathbf{K}_{s,a}(t))$, that is to say $\frac{N_{s,a}(t)}{N_s(t)} \geqslant \exp(-N_{s,a}(t)\mathbf{K}_{s,a}(t))$. Taking the logarithm and rearranging the terms, this condition rewrites $\mathbf{H}_{s,a}(t) \geqslant \log N_s(t)$ at each time step $t$. The action that is the closest to violate this condition or that violates this condition the most is the one of minimal IMED-RL index, $\arg\min_a \mathbf{H}_{s,a}(t)$, the one IMED-RL decides to play.

---

**Algorithm 1** IMED-RL: **I**ndexed **M**inimum **E**mpirical **D**ivergence for **R**einforcement **L**earning

---

**Require:** State-Action space $\mathcal{X}_{\mathbf{M}}$ of MDP $\mathbf{M}$, Assumptions 1, 2, 3
**Require:** Initial state $s_1$
   **for** $t \geqslant 1$ **do**
      Sample $a_t \in \arg\min\limits_{a \in \mathcal{A}_{s_t}} \mathbf{H}_{s,a}(t)$
   **end for**

---

Intuitions of the IMED-RL algorithm root to the control theory of MDPs and optimal bandit theory; IMED-RL intertwines the two and the regret proof exactly follows from the following intuitions.

**Control**  In control theory, we assume that both the expected rewards and transitions probabilities of an MDP $\mathbf{M}$ are known. Policy iteration (see Puterman [1994], Bertsekas and Shreve [1978]) is an algorithm that computes a sequence $(\pi_n)_n$ of deterministic policies that are increasingly strictly better until an optimal policy is reached. In the average-reward setting and under the ergodic assumption, a policy $\pi$ is strictly better than another policy $\pi'$ if $g_\pi(\mathbf{M}) > g_{\pi'}(\mathbf{M})$. The policy iteration algorithm computes the sequence of policies recursively in the following way. Initially, an arbitrary deterministic policy $\pi_0$ is chosen. At step $n + 1 \in \bar{\mathbb{N}}^*$, it computes $\mathbf{m}_{\pi_n}$ and $\mathbf{b}_{\pi_n}$ then swipes through the states $s \in \mathcal{S}$ in an arbitrary order until it reaches one state $s$ such that there exists $a \in \mathcal{A}(s)$ with $\mathbf{m}(s,a) + \mathbf{p}(\cdot|s,a)\mathbf{b}_{\pi_n} > \mathbf{m}_{\pi_n}(s) + \mathbf{p}_\pi(s)\mathbf{b}_{\pi_n}$. If such an $s$ does not exist, then it returns $\pi_n$ as an optimal policy. Otherwise, $\pi_{n+1}$ is defined as $\pi_{n+1}(s') = \pi_n(s')$ for all $s \neq s'$ and $\pi_{n+1}(s) \in \arg\max\{\mathbf{m}(s,a) + \mathbf{p}(\cdot|s,a)\mathbf{b}_{\pi_n}\}$. Such a step is called a policy improvement step. Policy iteration is guaranteed to finish in a finite number as the cardinal of $\Pi(\mathbf{M})$ is finite. At each step $n \in \mathbb{N}^*$, $\varphi_{\mathbf{M}(\pi_n)}$ is a local function that takes into account the whole dynamic of the MDP and allows to compute, *via* an *argmax*, an optimal choice of improvement (or optimal action) based on local information; $\varphi_{\mathbf{M}(\pi_n)}(\mathbf{r}(s,a) \otimes \mathbf{p}(\cdot|s,a)) = \mathbf{m}(s,a) + \mathbf{p}(s,a)\mathbf{b}_{\pi_n}$. IMED-RL uses $\varphi_{\widehat{\mathbf{M}}(\mathcal{A}(t))}$ and improves the skeleton similarly to policy iteration as it can be seen in the analysis 4.

**Bandit control**  A degenerate case of MDP would be one where there is only one state $s$ with $\varphi_{\mathbf{M}(\varphi)}(\mathbf{r}(s,a)) = \mathbf{m}(s,a)$ by choosing the bias function to be zero[6]. Playing optimally consists in playing an action with largest expected reward at each time step $t$, $a_t \in \arg\max_{a \in \mathcal{A}_s} \mathbf{m}(s,a)$.

**Bandit**  Learning occurs when rewards are unknown; this is the bandit problem. In that case, a lower bound on the regret similar to 1 exists. Under some assumptions on the reward distributions, optimal algorithms whose regret upper bounds asymptotically match the lower bound can derived. IMED Honda and Takemura [2015], KL-UCB Maillard et al. [2011], Cappé et al. [2013] are two such examples that use indexes, *i.e.* computes a number $I_{s,a}(t)$ at each time step and play $a_t \in \arg\min I_{s,a}(t)$. Such indexes are crafted to correctly handle the *exploration-exploitation* trade-off.

**RL in Ergodic MDPs**  The delayed rewards caused by the dynamic of the system is the main source of difficulty arising from having more than one state. IMED-RL combines control and bandit theory

---

[6]recall that the bias function is defined up to an additive constant

in the following way. At each time step $t$, a restricted MDP $\widehat{\mathbf{M}}_t(\mathcal{A}(t))$ is built from the empirical one $\widehat{\mathbf{M}}_t$. If the condition to belong to the skeleton is selective enough, then the potentials on the restricted empirical MDP $\widehat{\mathbf{M}}_t(\mathcal{A}(t))$ may become close to those of the restricted true MDP $\mathbf{M}(\mathcal{A}(t))$, that is $\|\varphi_{\widehat{\mathbf{M}}_t(\mathcal{A}(t))} - \varphi_{\mathbf{M}(\mathcal{A}(t))}\|_\infty$ is small. We want to make policy improvements by finding, at each state $s$ an action $a' \in \arg\max \varphi_{\mathbf{M}(\mathcal{A}(t))}(\mathbf{r}(s,a) \otimes \mathbf{p}(\cdot|s,a))$, play it enough that it belongs to the skeleton which will modify $\varphi$ and repeat until $\varphi_{\mathbf{M}(\mathcal{A}(t))} = \varphi_{\mathbf{M}}$. Using $\varphi$, the global dynamic is reduced to a local function so that at each state, the agent is presented a bandit problem. This bandit problem is well estimated if $\|\varphi_{\widehat{\mathbf{M}}_t(\mathcal{A}(t))} - \varphi_{\mathbf{M}(\mathcal{A}(t))}\|_\infty$ is small. As opposed to the control setting, the learning agent cannot choose the state in which to make the policy improvement step and it may be possible that no policy improvement step is possible at state $s_t$. However, thanks to the ergodic assumption 2 the agent is guaranteed to visit such a state in finite time, if it exists. There is a trade-off between the adptativity of the skeleton, *i.e.* how quickly one can add an improving action to define a new $\varphi$, and concentration of statistical quantities defined on the restricted MDP.

**Related work**  Our notion of skeleton is built on the work of Burnetas and Katehakis [1997]. We improve on their original notion of skeleton by correcting some troubles happening in the small sample regime. In particular, this forces the authors to introduce some forcing mechanism. The issues of the original definition and improvement induced by ours are listed in Appendix G. One key point of our definition is that the skeleton is defined using only empirical quantities, the number of samples, and does not depends on some arbitrary reference, such as the absolute time.

## 4  Regret of `IMED-RL`

In this section we state the main theoretical result of this paper, which consists in the `IMED-RL` regret upper bound. We then sketch a few key ingredients of the proof.

**Theorem 2** (Regret upper bound for Ergodic MDPs). *Let* $\mathbf{M} = (\mathcal{S}, \mathcal{A}, \mathbf{p}, \mathbf{r})$ *be an MDP satisfying assumptions 1, 2, 3. Let* $0 < \varepsilon \leqslant \frac{1}{3} \min\limits_{\pi \in \Pi(\mathbf{M})} \min\limits_{(s,a) \in \mathcal{X}_{\mathbf{M}}} \{|\Delta_{s,a}(\mathbf{M}(\pi))| : |\Delta_{s,a}(\mathbf{M}(\pi))| > 0\}$. *The regret of* `IMED-RL` *is upper bounded,*

$$\mathcal{R}_{\textit{IMED-RL}}(\mathbf{M}, T) \leqslant \left( \sum_{(s,a) \in \mathcal{C}(\mathbf{M})} \frac{\Delta_{s,a}(\mathbf{M})}{\underline{\mathbf{K}}_{s,a}(\mathbf{M}) - \varepsilon \Gamma_s(\mathbf{M})} \right) \log T + O(1), \tag{14}$$

*where* $\Gamma_s(\mathbf{M})$ *is constant that depends on the MDP* $\mathbf{M}$ *and state* $s$; *it is made explicit in the proof detailed in Appendix D. A Taylor expansion allows to write the regret upper bound as*

$$\mathcal{R}_{\textit{IMED-RL}}(\mathbf{M}, T) \leqslant \left( \sum_{(s,a) \in \mathcal{C}(\mathbf{M})} \frac{\Delta_{s,a}(\mathbf{M})}{\underline{\mathbf{K}}_{s,a}(\mathbf{M})} \right) \log T + O\left( (\log T)^{10/11} \right). \tag{15}$$

*Were the semi-bounded reward assumption changed to a bounded reward one with known upper and lower bound, the* $O\left( (\log T)^{10/11} \right)$ *could be made a* $O(1)$ *as explained in Appendix E.*

**Theorem 3** (Asymptotic Optimality). `IMED-RL` *is asymptotically optimal, that is,*

$$\lim_{T \to +\infty} \frac{\mathcal{R}_{\textit{IMED-RL}}(\mathbf{M}, T)}{\log T} \leqslant \sum_{(s,a) \in \mathcal{C}(\mathbf{M})} \frac{\Delta_{s,a}(\mathbf{M})}{\underline{\mathbf{K}}_{s,a}(\mathbf{M})}. \tag{16}$$

The proof of Theorem 3 is immediate from Theorem 2 by first dividing Equation 14 by $\log T$, then by taking the limit $T \to \infty$, and finally taking the limit $\varepsilon \to 0$.

**Remark**  While the regret *lower bound*, Theorem 1, is asymptotic by nature, our main Theorem 2 states a finite time *upper bound* on the regret of `IMED-RL`. Indeed, both Equations 14 and 15 are valid for all time $T$. The term $O(1)$ appearing in Equation 14 does not depend on time $T$ and is a constant that depends on both the MDP $\mathbf{M}$ and $\varepsilon$. This dependency is hard to be made explicit as this term is computed as limits of convergent series that are derived in the proof, see Appendix D. In Equation 14, the constant $\sum_{(s,a) \in \mathcal{C}(\mathbf{M})} \frac{\Delta_{s,a}(\mathbf{M})}{\underline{\mathbf{K}}_{s,a}(\mathbf{M}) - \varepsilon \Gamma_s(\mathbf{M})}$ in front of $\log T$ does not exactly match the asymptotic

upper bound $\sum_{(s,a)\in\mathcal{C}(\mathbf{M})}\frac{\Delta_{s,a}(\mathbf{M})}{\underline{\mathbf{K}}_{s,a}(\mathbf{M})}$ because of the $\varepsilon$-term in the denominators. Equation 15 states that using a bounded reward hypothesis, instead of semi-bounded, allows the constant in front of the leading $\log T$ term to exactly match the asymptotic one, even in the finite time regret upper bound. In both cases, Theorem 3 states that asymptotic optimality is achieved.

This Theorem proves the optimality of `IMED-RL` since the upper bound on the regret matches the lower bound of Theorem 1. Such a bound was asymptotically matched by the algorithm proposed by Burnetas and Katehakis [1997] and we recall that this algorithm and its problems are discussed in Appendix G. On the other hand, the current state-of-the-art algorithms `UCRL3` and `PSRL`, while having some theoretical guarantees, have not been proved to match the regret lower bound. On the practical side, Q-learning is often used without much theoretical guarantee because of its usually strong practical performances. In the experiments, we will compare `IMED-RL` to those three algorithms.

**Related work**  Theorems 2 and 3 prove that `IMED-RL` is achieving the optimal rate of exploration (in the exploitation-exploration tradeoff sense) for ergodic MDPs. Its theoretical guarantees are problem-dependent rather than worst-case/min-max. Comparing to the $\log T$ bound derived for `UCRL` in Theorem 4 of Jaksch et al. [2010], less known than the $\sqrt{T}$ bound, shows the benefit of our analysis for each instance, as we improve the constant factors in the leading terms: their dependency is $34D^2S^2A/\Delta$, where $\Delta$ is a sub-optimality gap and $D$ the diameter of the MDP.

**Sketch of proof**  Though a full proof is given in Appendix D, we sketch here the main proof ideas that follow directly from the intuitions behind the `IMED-RL` conception. The regret is decomposed into two terms, the **bandit** term when the local bandit problems defined by $\varphi_{\widehat{\mathbf{M}}_t(\mathcal{A}(t))}$ is well estimated, and the **skeleton improvement** term that controls the probability that the local bandit problem is not well estimated. This second term is managed by controlling the number of policy improvement steps and using concentration properties of empirical quantities defined on the skeleton.

The main Theorem 2 follows from the following proposition that is proved in Appendix D. Recall from Lemma 1 that for all state-action pair $x \in \mathcal{X}_{\mathbf{M}}$, $N_x(T) = \sum_{t=1}^{T} \mathbb{1}\{(s_t, a_t) = x\}$ counts the number of time the state-action pair $x$ has been sampled.

**Proposition 1.** *For all state-action pair $x \in \mathcal{X}_{\mathbf{M}}$, for all $\varepsilon > 0$,*

$$N_x(t) \leqslant B_x(T) + S(T), \tag{17}$$

*where we introduced the bandit term, $B_x(T)$, and the skeleton improvement term, $S(T)$,*

$$B_x(T) = \sum_{t=1}^{T} \mathbb{1}\left\{x_t = x, \mathcal{O}\left(\widehat{\mathbf{M}}_t\left(\mathcal{A}(t)\right)\right) \subseteq \mathcal{O}\left(\mathbf{M}\right), \|\mathbf{b}^{\widehat{\mathbf{M}}_t(\mathcal{A}(t))} - \mathbf{b}^{\mathbf{M}}\|_{\infty} \leqslant \varepsilon\right\}, \tag{18}$$

$$S(T) = \sum_{t=1}^{T} \mathbb{1}\left\{\overline{\mathcal{O}\left(\widehat{\mathbf{M}}_t\left(\mathcal{A}(t)\right)\right) \subseteq \mathcal{O}\left(\mathbf{M}\right), \|\mathbf{b}^{\widehat{\mathbf{M}}_t(\mathcal{A}(t))} - \mathbf{b}^{\mathbf{M}}\|_{\infty} \leqslant \varepsilon}\right\}. \tag{19}$$

*Furthermore, $\mathbb{E}\left(S(T)\right) = O(1)$, $\mathbb{E}\left(B_x(T)\right) = O(1)$ for a non-critical state-action pair, while for a critical state-action pair $x$,*

$$\mathbb{E}\left(B_x(T)\right) \leqslant \frac{\Delta_x\left(\mathbf{M}\right)}{\underline{\mathbf{K}}_x\left(\mathbf{M}\right) - \varepsilon\Gamma_s\left(\mathbf{M}\right)} \log T + O(1)$$

## 5  Numerical experiments

In this section, we discuss the practical implementation and numerical aspects of `IMED-RL` and extend the discussion in Appendix F. Source code is available on github[7].

**Computing `IMED-RL` index**  At each time step, we run the value iteration algorithm on $\widehat{\mathbf{M}}_t(\mathcal{A}(t))$ to compute the optimal bias and the associated potential function $\varphi_{\widehat{\mathbf{M}}_t(\mathcal{A}(t))}$. This task is standard. Once done, one must compute the value of the optimization problem $\mathbf{K}_{s,a}(t)$ which belongs to the category of convex optimization problem with linear constraint. Such problems have been studied

---

[7]Plain text URL is https://github.com/fabienpesquerel/IMED-RL

under the name of *partially-finite convex optimization*, *e.g.* in Borwein and Lewis [1991]. It is possible to compute $\mathbf{K}_{s,a}(t)$ by considering the Legendre-Fenchel dual and one does not need to compute the optimal distribution to know the value of the optimization problem.

**Proposition 2** (Index computation, Honda and Takemura [2015] Theorem 2). *Let $(s,a)$ be in $\mathcal{X}_{\mathbf{M}}$, $M = m_{max}(s,a) + \max\limits_{s' \in \mathcal{S}} \mathbf{b}^{\mathbf{M}}(s)$, and $\gamma > \varphi_{\mathbf{M}}(\mathbf{r}(s,a) \otimes \mathbf{p}(\cdot|s,a))$, then*

$$\underline{\mathbf{K}}_{s,a}(\mathbf{M}, \gamma) = \begin{cases} \max\limits_{0 \leqslant x \leqslant \frac{1}{M-\gamma}} \mathbb{E}_{\substack{R \sim \mathbf{r}(s,a) \\ S \sim \mathbf{p}(\cdot|s,a)}} \left[ \log\left(1 - \left(R + \mathbf{b}^{\mathbf{M}}(S) - \gamma\right)x\right)\right] & \text{if } M > \gamma \\ +\infty & \text{otherwise} \end{cases}. \quad (20)$$

*If $\gamma \leqslant \varphi_{\mathbf{M}}(\mathbf{r}(s,a) \otimes \mathbf{p}(\cdot|s,a))$, then $\underline{\mathbf{K}}_{s,a}(\mathbf{M}, \gamma) = 0$.*

In particular, this Proposition 2 sometimes allows to write $\mathbf{K}_{s,a}(t)$ almost in close form, *e.g.* when $\mathcal{F}_{s,a}$ defined in Asumptions 3 is a set of multinomials with unknown support (and only the upper bound $m_{max}$ is known). In Appendix F, we discuss this numerical computation further.

**Computational complexity**    In terms of state and actions spaces sizes, the complexity of `IMED-RL` at each time step scales as $O(S^2A)$, the complexity of value iteration. Indeed, at each time step, `IMED-RL` runs value iteration using actions available in the skeleton, then computes the indexes of the available actions at the current state, and finally pick an *argmin*. The complexity of value iteration is $O(S^2A)$, the complexity of computing the $A$ necessary indexes is $O(SA)$, and the complexity of picking an *argmin* amongst those $A$ indexes is $O(A)$. Therefore, the per-time-step complexity of `IMED-RL` scales as $O(S^2A)$. However, this scaling is mainly an upper-bound as value iteration is run with actions that are within the skeleton. By design of the skeleton, we experimentally observe that, after some time, the skeleton contains one action per state (the optimal one). We provide more details in Appendix F, *Lazy update* paragraph.

**Practical comparison**    In practice, most of the complexity of `IMED-RL` is in the analysis rather than in the algorithm: compared to `PSRL` and `UCRL3`, `IMED-RL` does not take a confidence parameter nor any hyperparameter. Also, `IMED-RL` uses value iteration as a routine, which is faster than the extended value iteration used in `UCRL3`. Q-learning technically takes an exploration parameter ($\varepsilon$-greedy exploration) or exploration scheme when it is slowly decreased with time.

**Environments**    In different environments, we illustrate in Figure 2 and Figure 3 the performance of `IMED-RL` against the strategies `UCRL3` Bourel et al. [2020], `PSRL` Osband et al. [2013] and Q-learning (run with discount $\gamma = 0.99$ and optimistic initialization). As stated during the introduction, any finite communicating MDP can be turned into an ergodic one, since on such MDPs, any stochastic policy $\pi : \mathcal{S} \to \mathcal{P}(\mathcal{A}_s)$ with full support $\text{Supp}(\pi(s)) = \mathcal{A}_s$ is ergodic. Hence by mixing its transition $\mathbf{p}$ with that obtained from playing a uniform policy, formally $\mathbf{p}_\varepsilon(\cdot|s,a) = (1-\varepsilon)\mathbf{p}(\cdot|s,a) + \varepsilon \sum\limits_{a' \in \mathcal{A}_s} \mathbf{p}(\cdot|s,a')/|\mathcal{A}_s|$, for an arbitrarily small $\varepsilon > 0$ one obtain an ergodic MDP.
In the experiments, we consider an ergodic version of the classical $n$-state river-swim environment, 2-room and 4-room with $\varepsilon = 10^{-3}$, and classical communicating versions ($\varepsilon = 0$).

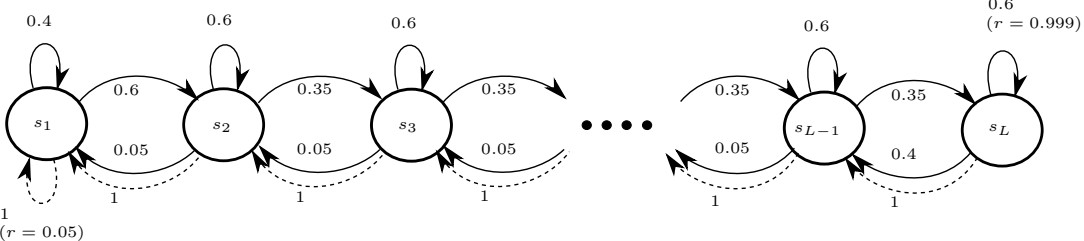

Figure 1: The ergodic $n$-state *RiverSwim* MDP. In each of the $n$ states, there are two actions `RIGHT` and `LEFT`. The left action is represented with a dashed line and the `RIGHT` with plain line. Rewards are located at the extremities of the MDP.

**n-states *RiverSwim* environment**    As illustrated by Figure 2, the performances of `IMED-RL` are particularly good and the regret of `IMED-RL` is below the regrets of all its competitors, even when the MDP is communicating only. This numerical performance grounds numerically the previous theoretical analysis. While using `IMED-RL` in communicating MDPs is not endorsed by our theoretically analysis, it is interesting to see how much this hypothesis amounts in the numerical performances of `IMED-RL`. We therefore ran an experiment on another classical environment, 2-rooms.

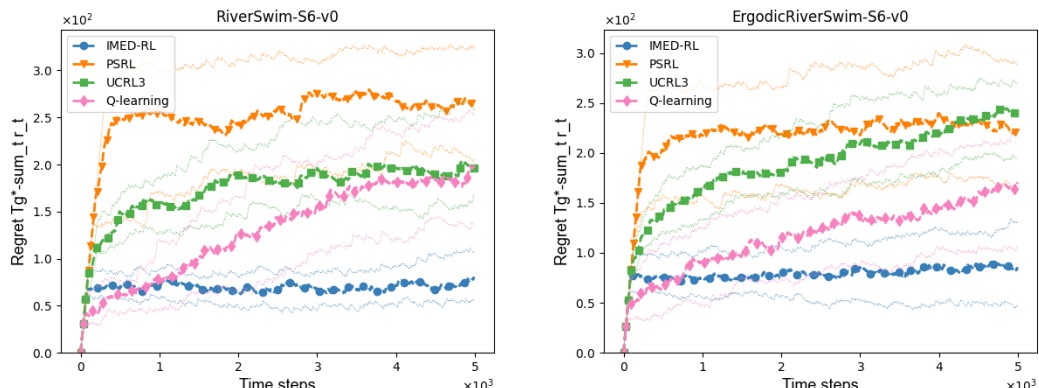

Figure 2: Average regret and quantiles (0.1 and 0.9) curves of algorithms on a standard communicating 6-states RiverSwim (left) and an ergodic 6-states RiverSwim (right).

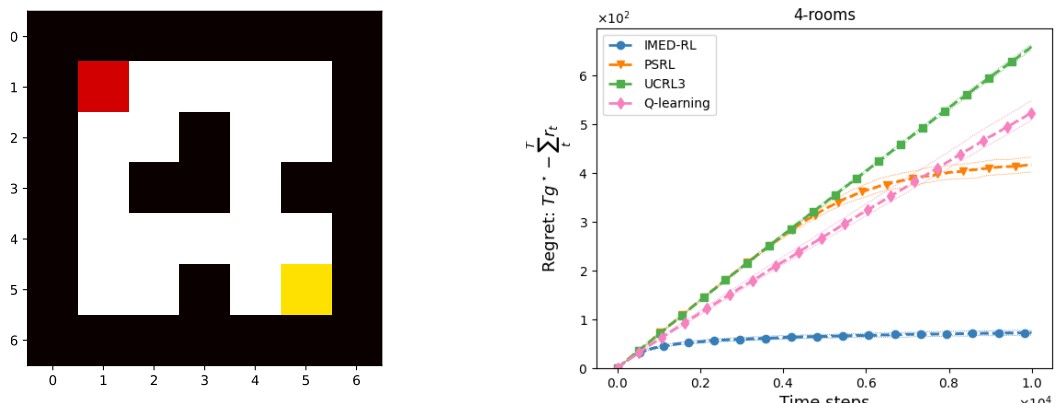

Figure 3: Average regret and quantiles (0.1 and 0.9) curves of algorithms (right) corresponding to learning on a 4-room (left) grid-world environment, with 20 states: the starting state is shown in red, and the rewarding state is shown in yellow. From the yellow state, all actions bring the learner to the red state. Other transitions are noisy as in a *frozen-lake* environment.

**n-rooms environment**    As illustrated by Figure 3, the performances of `IMED-RL` are particularly good, even surprisingly good, in this communicating only environment. Those experiments are a clue that the `IMED-RL` strategy may still be reasonable, although not necessarily optimal in some communicating MDPs. All experiments take less than an hour to run on a standard CPU.

**Future work**    Although not intended for non-ergodic MDPs, `IMED-RL` exhibits state-of-the-art numerical performances in communicating only MDPs (see Appendix F.2 for additional experiments). Hence, `IMED-RL` might prove to be insightful to pave the way towards the communicating case as it seems possible to get a controlled regret also in the case of communicating MDPs. Both the problem-dependent and worst-case regret bounds are interesting in this regard. Another direction we intend to explore is the adaptation of `IMED-RL` main ideas to *function approximation* frameworks, such as neural networks and kernel methods.

## Conclusion

In this paper, we introduced `IMED-RL`, a numerically efficient algorithm to solve the average-reward criterion problem under the ergodic assumption for which we derive an upper bound on the regret matching the known regret lower bound. Further, its surprisingly good numerical performances in communicating only MDPs open the path to future work in MDPs that are communicating only.

## Acknowledgments and Disclosure of Funding

This work has been supported by the French Ministry of Higher Education and Research, Inria, Scool, the Hauts-de-France region, the MEL and the I-Site ULNE regarding project R-PILOTE-19-004-APPRENF. The PhD of Fabien Pesquerel is supported by a grant from École Normale Supérieure.

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
