# OpenReview forum: "IMED-RL: Regret optimal learning of ergodic Markov decision processes"
_NeurIPS.cc/2022/Conference — NeurIPS 2022 Accept_

### Official Review · Reviewer_UdXd · 2022-07-10

**Rating:** 7
**Confidence:** 3
**Soundness:** 4 excellent
**Presentation:** 3 good
**Contribution:** 3 good

**Summary:**

This paper considers regret minimization in ergodic undiscounted MDPs under the average-reward criterion. It builds upon the classic results by Burnetas and Katehakis (1997) for this setting, and provides an adaptation of the IMED bandit algorithm (Honda and Takemura, 2015) to this setting. This algorithm, called IMED-RL, matches asymptotically the lower bound by Burnetas and Katehakis, and can thus be considered optimal. The theoretical results are complemented by numerical simulations showing that IMED-RL is very efficient in practice.

**Questions:**

1. How would the worst-case regret of IMED-RL compare with the one of UCRL and PSRL?
2. What do you think is the advantage of IMDEL-RL over the other exploration algorithms in practice?

Minor:
- line 19: communicative -> communicating
- Definition 1: s_1 should be s or vice-versa
- line 92: why is UCB "infamous"?
- line 124: communication -> communicating
- Footnote 2: square bracket
- line 258: "numerical issues" is an odd choice of words, it usually refers to numerical problems in the implementation. I think you mean something like "empirical aspects"
- line 273: missing space after PSRL

**Limitations:**

Limitations should be discussed more. For instance, how does IMED-RL scale to problems with many states and actions?

**Strengths And Weaknesses:**

The paper is well written and clear. The authors do a great job in explaining the theoretical framework, which is classic but is also a notoriously complicated one. The difference between existing results and original contributions is also well explained. I would only spend more words on describing the original IMED (bandit) algorithm of which IMED-RL is an adaptation to MDPs.

Originality is limited, since the work builds upon a well established theoretical framework and an existing algorithmic idea. However, there is some originality in the theoretical analysis, for instance in how the concept of skeleton.

The results are significant for the area of efficient exploration in RL. As mentioned by the authors, it is not easy to match the lower bound by B&K with a reasonably practical algorithm. Instead, besides its strong theoretical guarantees, IMED-RL shows good performance in practice too.

---

> ### Author Response · Authors · 2022-08-02
> **Response to reviewer UdXd**
>
> We thank you for your kind words regarding clarity, quality of explanations of the framework and difference to existing work, and for acknowledging the competitive empirical performance and significance of our results for the area of efficient exploration in RL.
> We agree to spend more words on describing the original IMED bandit algorithm, using the additional page available in the final version.
> We also thank you for your careful reading and cacthing the typos and language issues that we will correct in the CR version of the paper.
>
> Please consider reading the **general comment** where we adress two points that were raised by all the reviewers: worst case bounds and ergodic assumptions.
>
> ## Worst-case regret of IMED-RL
> Let us first restate that the focus of this work is on optimal problem-dependent regret bounds, not worst-case bounds.
> We believe there has already been a lot of works on worst-case bounds, and problem-dependent bounds have been somewhat overlooked.
> We intend to fill this gap in the literature.
>
> Comparing to the $\log(T)$ bound derived for UCRL in their paper (Theorem 4 of  [Jaksh et al, 2010], less known than the $\sqrt T$ bound) shows the benefit of our analysis for each instance,
> as we improve the constant factors in the leading terms (their dependency is $34^2S^2A/\Delta$, where $\Delta$ is a sub-optimality gap).
> Hence, we believe a similar improvement may be derived for the worst case, at least for "ergodic enough" MDPs.
> Due to this, we expect IMED-RL worst-case might be as good as the worst-case regret of UCRL and PSRL.
> It is difficult to say more without adding some proof and statement, though.
>
> Furthermore, the worst-case bound of UCRL and PSRL are stated for communicating MDPs whereas our work purposely focus on ergodic MDPs, as we target optimality. Also note that neither UCRL and PSRL are worst case optimal. To carefully derive a worst-case bound for IMED-RL, one would have to either prove some new guarantee of IMED-RL or prove a worst-case bound for "ergodic enough" MDPs.
>
> ## Advantage of IMED-RL
>
> IMED-RL is provably achieving the optimal rate of exploration (in the exploitation-exploration tradeoff sense) for ergodic MDPs. Its theoretical guarantees are problem dependent rather than worst-case.
>
> In practice, this property is supported by strong experimental performances against other state-of-the-art algorithm, with good performances even in non-ergodic environments. For instance, see Figure 3 (l. 287) and Fig. 5 (l. 960) where IMED-RL strongly outperforms PSRL, UCRL3 and Q-learning in an environment where efficient exploration is necessary due to the scarcity of the rewards. Other gridworld environments are presented Fig. 6 (l. 970) and Fig. 7 (l. 974) where IMED-RL is again the strongest competitor.
>
> From a practical view-point, we reported in Table 2 (l. 971) the average runtime of used algorithms and see that IMED-RL, while not the fastest still beats UCRL3 in speed.
>
> In practice, most of the complexity of IMED-RL is in the analysis rather than in the algorithm: Compared to PSRL and UCRL, IMED-RL does not take a confidence parameter (denoted $\delta$ in Fig. 1 of [Jaksh et al, 2010] and $\tau$ in section 3 of [Osband et al., 2013]). Also, IMED-RL uses value iteration as a routine, which is faster than the extended value iteration used in UCRL.
>
> We believe these are interesting features in favor of IMED-RL.
>
> ## Many states and actions - computational complexity
>
> We agree to expend the discussion on computational complexity in the main paper, using the additional page allowed.
>
> IMED-RL uses value iteration as a routine and the indexes are computed by solving (the dual of) a convex optimisation problem (Lemma 12, l. 909).
> The per-time-step complexity of IMED-RL is dominated by the complexity of value iteration.
> Indeed, at each time step, IMED-RL runs value iteration using actions available in the skeleton, then computes the $A$ indexes of the $A$ available actions at the current state, and finally pick an argmin.
> The complexity of value iteration is $O(S^2A)$, the complexity of computing the necessary indexes is $O(AS)$, and the complexity of picking an argmin is $O(A)$. The complexity of comuting an index is $O(S)$ because of the scalar product appearing in Lemma 12, l. 909.
> Therefore, the per-time-step complexity of IMED-RL scales as a $O(S^2A)$.
>
> Now, this scaling is mainly an upper-bound as value iteration is run with actions that are within the skeleton. By design of the skeleton, we experimentally observe that, after some time, the skeleton contains one action per state (the optimal one). We provide more details in the *Lazy update* paragraph l.926-937.
>
>
>  ### References
> see the General Comment

---

> > ### Comment · Reviewer_UdXd · 2022-08-08
> > **Thanks**
> >
> > Thanks you for your answer. In particular, I appreciated the clarification on problem dependent bounds and the comparison with UCRL. I encourage you to include this discussion in the final version of the paper.

---

> > > ### Author Response · Authors · 2022-08-09
> > > **Thanks for the response**
> > >
> > > We thank you again for reviewing our work and providing valuable comments that will help us to improve the final version of the paper.
> > >
> > > If there are no more concerns, we kindly request you to consider raising the score and/or increase your confidence in your judgement. We are happy to address any further questions you may have.

---

### Official Review · Reviewer_gX3h · 2022-07-11

**Rating:** 6
**Confidence:** 4
**Soundness:** 3 good
**Presentation:** 3 good
**Contribution:** 2 fair

**Summary:**

The paper considers ergodic MDPs and proposed an index policy IMED-RL. The IMED-RL index is built on the IMED policy for multi-armed stochastic bandits. Regret bound for IMED-RL is provided and shown to match the lower bound for ergodic RL problems.

**Questions:**

- It looks like one could compute the IMED-RL index for communicating MDPs. Would it be possible to apply IMED-RL to communicating MDPs with (possibly weaker) regret bounds?
- Are there any connection between the IMED-RL index and some optimistic values?

**Limitations:**

The assumptions and limitations are clearly stated in the paper.

**Strengths And Weaknesses:**

Strengths:
- The proposed policy is based on the IMED-RL index which is new for RL and different from commonly used optimism based  algorithms.

- Regret bounds of IMED-RL are provided for ergodic MDPs and the upper bound and lower bound match.

- Numerical experiments show better performance of IMED-RL compared with prior algorithms with certain regret guarantees.

Weaknesses:
- The ergodic assumption is very strong. Most MDPs including RiverSwim in the numerical experiment are not ergodic. The ergodic assumption is not typical and there are many RL algorithms with performance guarantees without the ergodic assumption. Results requiring the ergodic assumption seem very limited.

- The provided justification for focusing on ergodic MDPs is that it's the only class of MDPs with regret lower bounds. However, in Jaksch et al. [2010b], has a regret lower bound for the much more general class of communicating MDPs.

- Minor: the definitions of several notations are hard to find, making the paper not easy to follow. For examples, the definition of the count number N_{s,a}(T) is hidden in the statement of Lemma 1, and the definition of \phi_M is a bit hidden within lines.

---

> ### Author Response · Authors · 2022-08-02
> **Response to reviewer gX3h**
>
> Thank you for the suggestions regarding notations. This will be improved (using part of the additional space allowed for the CR version.). We will also remind the meaning or definition of $N_{sa} (T)$ in the Proposition 1, l. 253 to improve readability.
>
> Please consider reading the **general comment** where we adress two points that were raised by all the reviewers: worst case bounds and ergodic assumptions.
>
> ## Minimax lower bounds
> The lower bound for communicating MDPs you refer to from Jaksch et al. [2010b] is a minimax lower bound:
> the bound is obtained by building a specific nasty MDP that yields such regret, but do not say anthing about the achievable regret for a specific MDP.
> In contrast, problem-dependent lower bounds are defined for the MDP faced by the learner.
> We believe such bounds are valuable and more informative than minimax bounds: lower bounds have been helpful in bandits to derive state-of-the-art strategies such as KL-UCB or IMED.
> Such bounds are unfortunately not explicit for communicating, non-ergodic MDPs.
>
> ## Ergodic assumption
> We agree the ergodic assumption is indeed restrictive, and being more general is always more appealing, but there are reasons for this restriction:
> - We target exact asymptotic optimality, not just $\log (T)$ bound, not just $\sqrt T$ worst-case bound.
> - It is the only case in which explicit problem-dependent lower bounds are known and hence can be directly used to build a strategy.
> - Other UCRL related works also make this assumption, e.g. KL-UCRL modern analysis from [Talebi, Maillard 2018].
> - Existing proof strategies for state-of-the-art algorithms (UCRL, PSRL, etc) ensure a $O(\sqrt T)$ regret for communicating MDPs but fail to provide optimality guarantees (with $C \log(T)$ where C is the optimal constant) even in the ergodic case.
> - Deriving such a sharp result in the ergodic case is already non trivial and a contribution per se,
> and yet we believe it is insightful to pave the way towards the communicating case.
>
> Furthermore, the ergodic assumption is not as stringent as it looks.
> For instance in a communicating MDP, any epsilon-greedy policy, being a mixture with the random uniform policy, is ergodic.
> Hence, given the popularity of restricting to epsilon-greedy policies in RL, it is difficult to completely support the claim that the ergodic assumption is not typical.
> Also, putting a probability $10^{-15}$ to reach any state from any state makes any communicating MDP ergodic,
> and in our Theorem 2, this would only affect a constant hidden in the O(1) term in equation (13)
> (of course the constant blow up when ergodicity fails to hold).
> So, although we agree that ergodicity is restrictive, we believe in hindsight it is not that strong for the practitioner.
>
> Please remark that l.83-85, we highlight the fact that the ergodic assumption can be made on the set of MDPs or the set of policies. In our paper, we just made this assumption more visible than it usually is, for instance in $\epsilon$-greedy RL algorithms.
> We could rephrase our main result as: IMED-RL is an optimal learning strategy amongst epsilon-soft policies for communicating MDPs.
>
> ### Experiments
> Note that we do provide experiments both for standard and ergodic versions of Riverswim (Fig. 2, Fig. 4) and Gridworlds. We disagree we do not include experiments in the ergodic case.
>
> ## Weaker bounds in communicating MDPs
> Although not intended for non-ergodic MDPs, IMED-RL can indeed be applied to communicating MDPs.
> We actually provide many experiments showing the strategy is promising also in such cases
> (as we feel it is interesting to test the algorihm beyond its range of analysis).
> Hence, it seems IMED-RL might get a controlled regret also in the case of communicating MDPs.
> However, proving so is not easy and goes beyond the scope of this paper, whose primary goal is exact optimality.
> (see also answer *Extension to communicating MDPs* to reviewer mZe4).
> Yet, we have reasons to believe that in a class of communicating MDPs termed *"without bottleneck state to optimal recurrent classes"*, that captures most classical MDPs, IMED-RL stays order-optimal, hence have controlled regret (but this is speculative).
> We believe, as acknowledged by another reviewer, that our results are significant for the area of efficient exploration in RL, and also pave the way towards the communicating case.
>
> ## Link with optimism
> The IMED approach in bandits is based on likelihood ratio and information rather than optimism.
> It is actually more natural to interpret it as testing: if a seemingly suboptimal action can look good in a "confusing" environment that is the most similar to the empirically observed one.
> For an empirically suboptimal action, the test is made by comparing its "likelihood of being optimal" to its empirical frequency of sampling.
> We give an intuition of this approach l.178-186 of the paper.
> See [IMED] for further insights.
>
> ### References
> see the General Comment

---

> > ### Comment · Reviewer_gX3h · 2022-08-09
> > **After reading the response**
> >
> > Thank you for the clarification on the assumption and the answers to my questions. After reading the response, I still feel that restricting to ergodic MDPs makes the results too limited. One main challenge of exploration in RL is to efficiently find proper sequences of actions to reach certain import states that an epsilon-greedy policy can take exponential time to find. This challenge mostly disappears in ergodic MDPs since any policy can reach every state by any sequence of actions without exploration consideration. For general MDPs, one can indeed interpret the current results as providing an optimal epsilon-soft policy as the authors suggested in the response. But it's questionable whether the best epsilon-soft policy is meaningful for efficient exploration-exploitation trade-off since epsilon-soft policies are bound to have constant regret compared to the optimal non-epsilon-soft policy.
> >
> > Despite the current restriction, I think the IMED idea is very appealing and the analysis is solid. I feel like this can be a great work if in addition to the analysis on ergodic MDPs, the authors can provide some performance bounds (not necessarily optimal) for IMED-RL on more general MDPs to show that the potential benefit of IMED is not only limited to restricted settings.

---

> > > ### Author Response · Authors · 2022-08-09
> > > **Thanks for your input**
> > >
> > > Thank you again for acknowledging the ideas behind IMED-RL and the theoretical strength of the paper.
> > >
> > > We took your comment into consideration and tried to derive some performance bounds for the communicating only assumption.
> > > However, we designed IMED-RL for the ergodic case, and it really shows up in the analysis (and in the difficulty of modifying the proof without modifying the algorithm).
> > >
> > > While we agree it might be indeed possible to provide sub-optimal regret bounds for the more general communicating case, we also believe this would be at the cost of an unnecessarily complex analysis.
> > > Indeed, while a number of building blocks in the analysis extend almost as is beyond the ergodic case (e.g. control of bandit term appendix D.1, Proposition 8 ; control of equation (80) ; Lemma 5, 6 and 7), some key elements currently rely on the the fact that ALL states are visited (with high probability) a linear number of times by any policy followed by the algorithm (Lemma 4 and 3; the skeleton improvement term appendix D.2). In particular, Proposition 11 which prove an upper bound on the *expected time before a policy improvement* relies on the design of IMED-RL together with the ergodic assumption.
> > > We can try a few ideas in order to weaken this requirement, it turns out no-one is trivial (IMED-RL is intrinsically "local", and does not really take into account more global properties such as for instance the recurrence time to a state for the current optimal policy).
> > > IMED-RL mainly uses the bias function to handle the non-locality.
> > >
> > > In contrast, we believe that a small modification of IMED-RL could help extend the analysis beyond the ergodic case (still without being optimal for this extended case) in a simple way, targeting some recurrence of states.
> > > Hence, rather than a tedious analysis for IMED-RL, we believe changing IMED-RL to be more adapted to communicating  MDPs would be more interesting and worth.
> > > However, since such a modification is unnecessary for the ergodic case, we fear this would be confusing for the reader in the present work.
> > > Rather, such an extension can be a nice target for an extended journal version of this work. We agree to discuss such a possible perspective.
> > >
> > > Now, for the NeurIPS version, we believe the contribution should be focused on the ergodic case only.
> > > We take your input into consideration and will of course clarify these limitations and possible extensions.
> > > Taking into account the remark of the area chair GHFZ, we indeed agree to insist a bit more on the importance of the ergodic assumption in the MDP literature.

---

### Official Review · Reviewer_mZe4 · 2022-07-11

**Rating:** 6
**Confidence:** 3
**Soundness:** 3 good
**Presentation:** 3 good
**Contribution:** 3 good

**Summary:**

This paper studies regret minimization in infinite-horizon average-reward MDPs. They propose a new algorithm based on Indexed Minimum Empirical Divergence (IMED) bandit algorithm, and show that the new algorithm achieves a regret matching the asymptotic problem-dependent lower bounds.

**Questions:**

1. How should I compare the asymptotic problem-dependent lower bound with the $\sqrt{T}$-type minimax lower bound? Does the algorithm also ensures a $\sqrt{T}$ finite time regret?
2. What's the difficulty of extending current analysis to the communicating setting?
3. The regret definition is different from the one used in UCRL paper: $R_T = \sum_t (g_{\pi^{\star}} - r_t)$. How should I compare these two types of definitions?
4. Do you see a direct way to extend to linear function approximation?

**Limitations:**

See "Strengths And Weaknesses"

**Strengths And Weaknesses:**

Strength: the paper is clearly written. The proposed algorithm has strong theoretical guarantee and competitive empirical performance.

Weakness: Many important details are hidden in the Appendix (for example, the limitation of existing work is deferred to Appendix G). Thus, the contribution of this paper is not clear at first glance compared to existing works. I think more highlight s on the

---

> ### Author Response · Authors · 2022-08-02
> **Response to reviewer mZe4**
>
> Thank you for acknowledging the theoretical guarantees and empirical performances of IMED-RL. We appreciate that you found the paper clearly written.
> We agree to use the additional one page available in the CR version to improve the discussion regarding limitation of existing work and bring back part of existing work described in Appendix G to the main paper.
>
> Unfortunately, your *Weakness* paragraph ends up with a broken sentence. We are happy to discuss any further limitation you might want us to address.
>
> Please consider reading the **general comment** where we adress two points that were raised by all the reviewers: worst case bounds and ergodic assumptions.
>
> ## Finite-time regret
> Equation 13, l. 234 is actually valid for all time $T$: The term $O(1)$ is a constant that depends on both the MDP and $\epsilon$ but not on the time, see Equation 57, Prop. 8, l. 715, that expresses explicitly part of the term within the $O(1)$. We write it as $O(1)$ because it is the limit as $T$ tends to infinity of a convergent sequence, and to be coherent with the asymptotic lower-bound.
> Equation 14 shows a stronger result, namely exact optimality (no more $\epsilon$): This is made at the cost of a $O((\log T)^{10/11})$ in the general case, but only at the cost of a constant for bounded rewards support, e.g. in $[0,1]$, as stated line 237.
> However, the constants are difficult to explicit and handle. We agree to better explain this in the final version.
>
> ## Minimax
> Comparing to the $\log(T)$ bound derived for UCRL in their paper (Theorem 4 of  [Jaksh et al, 2010], less known than the $\sqrt T$ bound) shows the benefit of our analysis for each instance, as we improve the constant factors in the leading terms (their dependency is $34^2S^2A/\Delta$, where $\Delta$ is a sub-optimality gap).
> Hence, we believe a similar improvement may be derived for the worst case, at least for "ergodic enough" MDPs.
> Due to this, we expect IMED-RL worst-case might be as good as the worst-case regret of UCRL and PSRL.
> It is difficult to say more without adding some proof and statement, though.
>
> Furthermore, the worst-case bound of UCRL and PSRL are stated for communicating MDPs whereas our work purposely focus on ergodic MDPs, as we target optimality. Also note that neither UCRL and PSRL are worst case optimal. To carefully derive a worst-case bound for IMED-RL, one would have to either prove some new guarantee of IMED-RL or prove a worst-case bound for "ergodic enough" MDPs.
>
> ## Extension to communicating MDPs
> A difficulty in the analysis, based on the stochastic policy iteration intuition (l.187-228), is to ensure the algorithm visits fast enough a state in which a policy improvement step is possible.
> This is achieved easily in ergodic MDPs, but in general may require a more intricate analysis.
> Further, another difficulty is in the lower bounds: They are explicit only for the ergodic case, and otherwise involve a complicated optimization problem. This indicates that, as-is, IMED-RL is most certainly not optimal for generic non-ergodic MDPs and should be modified, but also that an optimal strategy for communicating, non-ergodic MDPs may be computationally challenging.
> Our experiments suggest that IMED-RL may still enjoy interesting regret bounds beyond the ergodic case.
>
> Given this, one can imagine one of the two things: extend the algorithm to the communicating case or prove that IMED-RL still enjoys some desirable theoretical guarantee, even in the communicating case. For instance, one could imagine that the regret is still logarithmic with a non-optimal constant in front of the logarithm. As such, IMED-RL would act as a computationally efficient relaxation of a difficult problem.
>
> ## Pseudo-regret
> In UCRL and related works, the authors indeed often use the pseudo-regret in lieu of the regret used in the lower bounds (Agrawal, Burnetas & Katehakis, Graves & Lai).
> The two quantities differ by replacing the T-steps cumulative reward of an optimal policy by T times the optimal gain $g^\star$.
> It is known at least from (Agrawal and Teneketzis, 1989) that the difference between the two is at most a constant in a finite-state communicating MDP.
> It is explicitely mentionned in the footnote 1 of [Jaksh et al, 2010] (p. 1565, just before section 1.1 where the authors introduce the regret).
>
> ## Linear function approximation
> From an algorithmic viewpoint, it seems natural to replace the value iteration procedure used to compute the empirical best policy with any one adapted to linear function approximation.
> Then one would also have to handle approximation of the counts $N_{sa} (t)$ (perhaps using ideas from [Ostrovski et al, 2017]).
> Now, any theoretical analysis of such extension seem hypothetic at the present time.
>
> ### References
> see the General Comment

---

### Author Response · Authors · 2022-08-02
**Thank you for your feedbacks**

# General Comment
We thank the reviewers for their kind words regarding clarity, quality of explanations of the framework and difference to existing work, originality in the approach and analysis, and for acknowledging the strong theoretical guarantee with a regret bound matching the lower bound, competitive empirical performance and significance of our results for the area of efficient exploration in RL.

While we adress specific questions in the answer to each reviewers, we would also like to emphasize the contribution of IMED-RL by adressing two points that were raised by all the reviewers: worst case bounds and ergodic assumptions.
## Worst case
We feel like many recent works focused on worst-case regret bounds only and studying problem-dependent optimal regret bounds has been somewhat overlooked.
Hence we are happy to contribute filling this gap in the literature.

The worst-case bound of UCRL and PSRL are stated for communicating MDPs whereas our work purposely focus on ergodic MDPs, as we target optimality. Also note that neither UCRL and PSRL are worst case optimal. To carefully derive a worst-case bound for IMED-RL, one would have to either prove some new guarantee of IMED-RL beyong ergodicity or prove a worst-case bound for "ergodic enough" MDPs. But even in this case, comparing to worst-case non-ergodic bounds seem little justified.

The lower bound for communicating MDPs from Jaksch et al. [2010b] is a minimax lower bound:
it is obtained by building a specific nasty MDP that yields such regret, but do not say anthing about the achievable regret for a specific MDP.
In contrast, problem-dependent lower bounds are defined for the MDP faced by the learner.
We believe such bounds are valuable and more informative than minimax bounds: lower bounds have been helpful in bandits to derive state-of-the-art strategies such as KL-UCB or IMED.
Such bounds are unfortunately not explicit for communicating, non-ergodic MDPs.

Comparing to the $\log(T)$ bound derived for UCRL in Theorem 4 of  [Jaksh et al, 2010], less known than the $\sqrt T$ bound, shows the benefit of our analysis for each instance, as we improve the constant factors in the leading terms (their dependency is $34^2S^2A/\Delta$, where $\Delta$ is a sub-optimality gap).
Hence, we believe a similar improvement may be derived for the worst case, at least for "ergodic enough" MDPs.
Due to this, we expect IMED-RL worst-case might be as good as the worst-case regret of UCRL and PSRL.
It is difficult to say more without adding some proof and statement, though.
## Ergodic assumption
While the ergodic assumption is indeed restrictive, and being more general is always more appealing, there are reasons for this restriction:
- We target exact asymptotic optimality, not just $\log (T)$ bound, not just $\sqrt T$ worst-case bound.
- It is the only case in which explicit problem-dependent lower bounds are known and hence can be directly used to build a strategy.
- Other UCRL related works also make this assumption, e.g. KL-UCRL modern analysis from [Talebi, Maillard 2018].
- Existing proof strategies for state-of-the-art algorithms (UCRL, PSRL, etc) ensure a $O(\sqrt T)$ regret for communicating MDPs but fail to provide optimality guarantees (with $C \log(T)$ where C is the optimal constant) even in the ergodic case.
- Deriving such a sharp result in the ergodic case is already non trivial and a contribution per se, and yet we believe it is insightful to pave the way towards the communicating case.

Please remark that l.83-85, we highlight the fact that the ergodic assumption can be made on the set of MDPs or the set of policies. In our paper, we made this assumption more visible than it usually is, e.g. in $\epsilon$-greedy RL algorithms.
For instance, in a communicating MDP, any epsilon-greedy policy, being a mixture with the random uniform policy, is ergodic.
Hence, given the popularity of restricting to epsilon-greedy policies in RL, we believe in hindsight it is not that strong for the practitioner and more typical than it may seem.

A take-home message is: For all positive epsilon, IMED-RL is an optimal learning strategy amongst epsilon-soft policies for communicating MDPs.
## References
Jaksch, T., Ortner, R., & Auer, P. (2010). Near-optimal Regret Bounds for Reinforcement Learning. In Journal of Machine Learning Research.
Honda, J., & Takemura, A. (2015). Non-Asymptotic Analysis of a New Bandit Algorithm for Semi-Bounded Rewards. In Journal of Machine Learning Research.
Talebi, M. S., & Maillard, O.-A. (2018). Variance-Aware Regret Bounds for Undiscounted Reinforcement Learning in MDPs, Algorithmic Learning Theory. PMLR.
Ostrovski, G., Bellemare, M. G., Oord, A., & Munos, R. (2017). Count-based exploration with neural density models. In International conference on machine learning. PMLR.
Osband, I., Russo, D., & Van Roy, B. (2013). (more) efficient reinforcement learning via posterior sampling. Advances in Neural Information Processing Systems, 26.

---

### Meta-Review · Area_Chair_GHFZ · 2022-09-08

**Recommendation:** Accept
**Confidence:** Certain

**Metareview:**

Regret theory for ergodic undiscounted infinite-horizon MDP was largely an open problem. The authors made an effort to fill in this gap. All reviewers see merits of the analysis, and the rebuttal has addressed most of the reviewers' concerns. Please make sure to incorporate necessary changes and add missing citations while preparing the final paper.

**Award:**

No

---

### Decision · Program_Chairs · 2022-09-14

Accept